# Geometric Morphometrics and Genetic Diversity Analysis of Chalcidoidea (*Diglyphus* and *Pachyneuron*) at Various Elevations

**DOI:** 10.3390/insects15070497

**Published:** 2024-07-03

**Authors:** Ouyan Xi, Shuli Zhang, Jinzhe Li, Hongying Hu, Ming Bai

**Affiliations:** 1College of Life Science and Technology, Xinjiang University, Urumqi 830017, China; xoy@stu.xju.edu.cn (O.X.); zhangsl_1998@163.com (S.Z.); lijinzhe990420@126.com (J.L.); 2Xinjiang Key Laboratory of Biological Resources and Genetic Engineering, Urumqi 830017, China; 3Institute of Zoology, Chinese Academy of Sciences, Beijing 100864, China; baim@ioz.ac.cn

**Keywords:** Eulophidae, Pteromalidae, COI barcoding, wing shape, haplotype

## Abstract

**Simple Summary:**

Parasitic wasps are natural enemies of numerous significant agricultural and forestry pests and therefore play an important role in biological control. With the development of integrative taxonomy, a greater variety of techniques are being applied to species identification. Therefore, integrated taxonomy was utilized in this study to identify parasitic wasps. Xinjiang is the largest province in China in terms of land area. However, the resources and environmental adaptation characteristics of Chalcidoidea have been little researched. In this study, we utilized geometric morphometrics and COI gene haplotype analysis and found that the morphology and genetic differentiation of Chalcidoidea changed at different altitudes.

**Abstract:**

Eulophidae and Pteromalidae are parasitic wasps with a global distribution and import for the biological control of pests. They can be distributed in different altitude regions, but their morphological and genetic adaptations to different altitudes are unclear. Here, we collected specimens that belong to Eulophidae and Pteromalidae from various altitudinal gradients, based on integrated taxonomic approaches to determine the species composition, and we analyzed their body shape and size from different altitudes using geometric morphometrics. Then, we performed an analysis of the *D. isaea* population’s haplotype genes to illustrate their genetic diversity. As a result, eight species that belong to two genera, *Diglyphus* Walker (Eulophidae) and *Pachyneuron* Walker (Pteromalidae), were identified, including two newly recorded species from China (*D. chabrias* and *D. sabulosus*). Through a geometric morphometrics analysis of body shape, we found that a narrow forewing shape and a widened thorax are the significant characteristics of adaptation to high-altitude environments in *D. isaea* and *P. aphidis*. Additionally, the body size studies showed a principal relationship between centroid size and altitude; the size of the forewings and thorax increases at higher altitudes. Next, using haplotype analysis, 32 haplotypes were found in seven geographic populations with high genetic diversity of this species. Our research provides preliminary evidence for the morphological and genetic diversity adaptation of parasitic wasps to extreme environments, and these data can provide important references for investigations on the ecological adaptability of parasitic wasps.

## 1. Introduction

Extreme habitat transitions, such as life in high mountains, deep oceans and caves, can drive changes in insect body size. Geometric morphometrics (GM) is a well-established and commonly used analytical framework that is extensively employed in several fields [1], for example, species identification [2,3,4,5], phylogeny [6] and environmental adaptation. The adaptive mechanisms of insects in extreme environments are a key issue in ecological research, including variations in body size and forewing area in hymenopteran species such as Apidae, Vespidae and Formicidae [7,8,9] and in other insect groups. Firstly, body size changes linked to community reorganization could have a significant impact on trophic interactions by causing modifications in the transfer and distribution of biomass, as well as on the stability of food webs and the functioning of ecosystems. For instance, some studies have been carried out on geometrid moths [10], damselflies [11], butterflies (*Vanessa carye*) [12] and Chalcids [13,14,15]. When there are trade-offs between characteristics, different selection forces may drive related groups towards different evolutionary optima [16]. Secondly, the wings of insects are a significant locomotor organ and play a crucial role in studying how insects adapt to their environment. The present study of wings is based the facts that the form and size of wings change as altitude and latitude increase in *Trilophidia annulate* (Orthoptera) [17] and that wing morphological differentiation was observed in *Apis cerana* (Hymenoptera) in different geographic regions [18]. However, fewer studies have focused on the adaptations of Chalcidoidea (Hymenoptera), which have a tiny body size and hidden living places.

The existence of large amounts of molecular and morphological variation within natural populations presents an evolutionary problem [19]. In general, the greater a species or population’s genetic diversity, the more adaptable it is to the environment, which makes it easier to explore new regions and improve its range. Changing climates are expected to have profound effects on the genetics of insect populations. Mitochondrial COI have also been used in population genetic studies, and genetic diversity helps species to adapt to unsuitable environments [20,21]. Little research has been carried out on the genetic diversity of parasitic wasps in Xinjiang. In this study, we utilized haplotype analysis of the mtCOI gene in order to understand the genetic diversity of parasitic wasps from different geographic populations.

The superfamily Chalcidoidea is the second largest group of Hymenoptera with more than 22,000 described species; they have been found in all zoogeographic zones of the globe. The majority of species live parasitically and can be parasitized at various stages of insect development; furthermore, several parasitic wasps have been successfully utilized for biological pest control [22]. *Diglyphus* Walker belongs to the family of Eulophidae, among which most of the species are important parasitic natural enemies; *D. isaea* is the primary parasitoid of agromyzid leafminers (Diptera) [23]. *Pachyneuron* Walker belongs to Pteromalidae, which is widely distributed in every zoogeographic region. Until now, 64 species have been described all over the world, among which 19 species are known from China [24,25]. Some species of *Pachyneuron* are parasitic wasps of many agricultural pests, such as pupal parasitoids of Diptera and Lepidoptera, while some species are hyperparasites of Aphididae [24,26]. For example, *Pachyneuron aphidis* is the dominant hyperparasitoid of *Aphidius gifuensis* (Hymenoptera), which is a key natural enemy of the green peach aphid *Myzus persieae* (Hemiptera) [27,28,29]. Biological control presupposes the accurate identification of the species. Nevertheless, its taxonomic classification is perplexing, and its physical structure is smaller and more comparable, leading to the application of morphology. Integrative taxonomy is the practice of using all available sources of data to identify the limits of species [30]. There are multiple methods available to define species, including molecular and morphological data [31,32].

Xinjiang is located in northwestern China, with various landscapes including extremely high mountainous areas such as the Eastern Pamir Plateau (average 4000 m), the Tianshan Mountains (average 4500 m) and the Kunlun Mountains (average 5500 m) [33,34]. This provides a suitable sampling area for parasitic wasp resources and environmental adaptation research. However, the resources of the two genera are still poorly known, especially in alpine habitats. Based on the investigation and classification of the genera *Diglyphus* and *Pachyneuron*, we found *D. isaea* and *P. aphidis* to be widely distributed at different altitudes. Secondly, we selected *D. isaea* and *P. aphidis* to explore changes in wing shape and thorax, as well as to clarify the morphological variations in response to extreme environments. Next, utilizing the COI haplotype data of *D. isaea*, we found a positive correlation between Fst and geographic distance; the key factor in genetic diversity was longitude. Finally, studying morphological adaptations in the wing and thorax of parasitic wasps will help us to understand their altitudinal adaptations. *D. isaea* haplotype analysis allows us to investigate the population’s genetic diversity. Our study enriches the resources on Chalcidoidea and allows us to understand the mechanism of these insects’ environmental adaptation at different altitudes.

## 2. Materials and Methods

### 2.1. Study Area and Collection of Samples

The specimens were collected from the southern parts of Xinjiang in China. The Eastern Pamir Plateau and the Altun Mountain National Nature Reserve (Appendix A) represented the high-altitude areas, where the average altitude of the sampling sites is about 2000 m. The remaining samples were obtained from the plain areas (Table 1).

A total of 507 specimens belonging to the genera *Diglyphus* and *Pachyneuron* were collected through net-sweeping and malaise traps in 2020 and 2023, and they were placed in anhydrous ethanol (99%) for further research. All the specimens were preserved in the Insect Collection of the College of Life Science and Technology, Xinjiang University, Urumqi, Xinjiang, China (ICXU).

### 2.2. Morphological Analysis

#### 2.2.1. Morphological Identification

Taxonomic identification of the morphology of the two genera (*Diglyphus* and *Pachyneuron*) was based on females. The specimens were sorted under a stereomicroscope (NOVEL JSZ6) and divided into genera followed by the keys of *Diglyphus* and *Pachyneuron* [35,36]. Then, their species were identified under a stereomicroscope (Nikon SM745T) by referring to the relevant literature [37,38,39,40].

#### 2.2.2. Morphometric Analysis

The right forewing of the specimen was removed from the wing base under a Nikon stereomicroscope. A total of 151 forewings of *Diglyphus* (*D.iseae* 42, *D. chabrias* 35, *D. sabulosus* 29, *D. albiscapus* 22, *D. crassinervis* 23) and 108 of *Pachyneuron* (*P. aphidis* 32, *P. grande* 46, *P. solitarium* 30) were obtained. Then, they were made into slide specimens to be photographed using Nikon microscope image acquisition software (Nikon Ci), all taken under 4× optics. Seven landmark sites were selected on the forewing (Figure 1). First, we used tpsUtil software (1.78) to convert the images to tps files. Then, tpsDig2 (2.31) was used to mark points and convert them to 2D coordinates [41]. In the end, generalized Procrustes analysis (GPA) was used to rotate, translate and scale the landmarks of all samples to eliminate the effects of non-morphological changes. To assess and test for within-group differences, we used principal component analysis (PCA) and canonical variate analysis (CVA) with MorphoJ 1.07a [42], and the Procrustes distance and Mahalanobis distance were determined during the CVA. The permutation test for pairwise distances all tested 10,000 simulations.

GM was utilized for cluster analysis. The Mahalanobis distance was constructed into a data matrix. PAST (4.0) parameters were utilized, and the UPGMA model was selected. In total, 1000 permutation runs were conducted.

### 2.3. Molecular Analysis

For the extraction of genomic DNA, the specimens were removed into PBS buffer and washed twice. Then, they were washed twice with distilled water, removed and dried on filter paper for moisture and set aside. The kit used for DNA extraction was the TIANamp Genomic DNA Kit (Beijing, China). The final DNA obtained was stored at −20 °C. PCR amplification of *Diglyphus* mtCOI [23] was carried out (primer sequences: COIS-F: 5′-TAAGATTTTGATTATTRCCWCC-3′; COI2613-R: 5′-ATTGCAAATACTGCACCTAT-3′). The total volume of the amplification COI reaction system was 25 μL, consisting of 12.5 μL PCR buffer master mix, 0.5 μL of each primer, 5 μL DNA and 6.5 μL ddH_2_O. After the response program of an initial denaturing step of 1 min at 94 °C, the following thermocycling conditions were applied: six cycles of 1 min at 94 °C, 1 min 30 s at 45 °C and 1 min 15 s at 72 °C, followed by a further 36 cycles of 1 min at 94 °C, 1 min 30 s at 51 °C and 1 min 15 s at 72 °C. There was a final extension of 10 min at 72 °C. *Pachyneuron* mtDNA COI primer sequences were FWPT-F: 5′-CCTGGTTCTTTRATTGGTAATGATC-3′; Lep-R: 5′-TAAACTTCTGGATGTCCAAAAA-3′ [43]. The response program was as follows: 4 min at 94 °C, then 35 cycles of 30 s at 94 °C, 30 s at 51 °C, 1 min at 72 °C and a final extension of 10 min at 72 °C. The amplification products were detected using 1% agarose gel electrophoresis and sequenced by the Bioengineering (Shanghai) company.

A total of 32 COI gene sequences were obtained from the eight species and uploaded to GenBank. Two sequences, *Necremnus tutae* (Eulophidae) and *Halticoptera longipetiolus* (Pteromalidae), were downloaded from NCBI as outgroups. Of these, 19 sequences belong to five species of *Diglyphus*, while 13 belong to three species *of Pachyneuron*. The obtained sequences were spliced in Bioedit software (7.2.5) and saved in ‘fas’ format. The obtained sequences were compared with Blast in NCBI (https://www.ncbi.nlm.nih.gov/; accessed on 15 March 2024) to determine the accuracy of the obtained sequences.

Then, Clustal W in MEGA X software (11.0.13) was used to perform pairwise and multiple alignment [44]. After the alignment was complete, the sequence was reduced at both ends and saved in FASTA format for subsequent tree building. Finally, Phylosuite software (v1.2.3) was used to construct the maximum likelihood tree (ML) [45]. For phylogeny, we employed Model Finder [46] in IQ-tree in all cases to automatically select the optimal substitution model [47]. A total of 1000 ultrafast bootstrap [48] replicates were performed to assess the robustness of the results. The resulting ML trees were visualized and annotated using iTOL [49].

ABGD automatically partitioned DNA barcode sequences based on the barcode gap using iterative model-based confidence limits for intraspecific divergence [50]. The resulting K2P distance [51] matrix was then analyzed using the ABGD online tool (https://bioinfo.mnhn.fr/abi/public/abgd/abgdweb.html; accessed on 29 March 2024) for partitioning. The relative gap width was set to 1.0, while other parameters were left at their default values.

### 2.4. Analysis of Morphology of the Two Genera at Different Altitudes

Differences in the morphology of *D. isaea* and *P. aphidis* were compared at different altitudes. High-altitude samples were mainly obtained from the Eastern Pamir Plateau and Altun Mountain National Nature Reserve, while low-altitude control groups were obtained from surrounding county towns in the southern region of Xinjiang.

The variations in elevational gradients were studied by comparing the forewings and thorax. The thorax of *D. isaea* was used in order to analyze the shape using 2D geometric morphometrics for the dorsal view, with 30 semi-landmarks. In the forewings, 3 landmarks and 40 semi-landmarks were used (Figure 2A,B). *P. aphidis* had 40 semi-landmarks on the thorax (dorsal view), and the forewings contained 4 landmarks and 30 semi-landmarks. These landmarks were digitized using tpsDig2 (Figure 2C,D).

Principal component analysis (PCA) and canonical variate analysis (CVA) were conducted on the normalized data using MorphoJ (1.07a). In order to observe and analyze the changes in forewing size at different altitudes, the ‘geomorph’ package of R was used to calculate the centroid size (CS) [52].

### 2.5. Analysis of D. isaea Haplotypes at Different Altitudes

*D. isaea* from different locations were collected separately to obtain 44 COI gene sequences (Appendix A). The number of haplotypes was counted using Dnasp 6.0 software. Climate data were obtained from Weather Data 24 h (https://www.tianqi24.com/; accessed on 10 May 2024) and the China Weather Data Center (http://data.cma.cn/; accessed on 2 May 2024). Elevation data were obtained in the field. The Mantel test was used in the Vegan package (R 4.3.2) to calculate the correlation of haplotype diversity *Hd* and nucleotide diversity *Pi* with elevation and climatic data [53]. The haplotype network was constructed based on the number of haplotypes using the Median Joining Network method in PopART software (1.7).

Fst measures the amount of genetic variance explained by group structure based on Wright’s F-statistics (Fst) [54]. Our results were calculated using DnaSP software, along with a matrix of straight-line geographic distances between sample points of each population. The Mantel test was used to calculate the correlation (*p*-value) between genetic and geographic distances. We used IBD v.1.53 (Isolation by Distance) software, and 1000 replicate samples were run to test for significance [55].

## 3. Results

### 3.1. Morphological Identification

#### 3.1.1. Morphological Classification

In this study, we examined 507 specimens of two genera and eight species. Of these, five species are in *Diglyphus* and three are in *Pachyneuron*. According to the field collection and specimen identification, these two genera have a large number of specimens with a wide distribution, which is suitable for subsequent research. Eight species were identified through morphological characteristics and the COI gene, among which five species are in *Diglyphus* and three species are in *Pachyneuron* (Appendix A). In *Diglyphus*, *D. chabrias* and *D. sabulosus* are newly reported from China. We found that *D. isaea* and *P. aphidis* were abundant and had a broad distribution. All morphological data are shown in Appendix A.

#### 3.1.2. Classification of Geometric Morphometrics

From the principal component analysis (PCA) results of the forewings, it was found that the first two principal components accounted for 42.76% and 24.07%, respectively, of the total variation in the 108 *Pachyneuron* specimens. The first two principal components were plotted to indicate variation along the two axes, with approximately 90% of the specimen points included in the 90% equal-frequency ellipses of the plots. The PCA results revealed a partial overlap between *P. aphidis* and *P. solitarium*, while *P. grande* was completely differentiated from them (Figure 3B). Morphological views show that *P. aphidis* and *P. solitarium* are more similar, while *P. grande* is also larger and has a wing pattern that differs markedly from that of the other two species (Appendix A).

In the canonical variate analysis (CVA) of *Pachyneuron* forewings, CV1 and CV2 were used as axes to construct the scatterplot. The results show a clear division between the three species (Figure 3A). The Mahalanobis and Procrustes distances between the groups were measured. The pairwise cross-validated reclassification test was efficient for separating the specimens. There are significant differences (*p* < 0.0001) in the *p*-values from 10,000 permutation tests for the Mahalanobis distances between groups, which indicates that the results of this study are statistically significant.

In the CVA of 151 specimens from *Diglyphus*, CV1 and CV2 accounted for 70.46% and 26.52% of the total variance. Scatterplots were plotted using CV1 and CV2, and it was found that *D. isaea* was better delineated from the other four species, whereas *D. sabulosus* and *D. albiscapus* partially overlapped, and there was also a small overlap between *D. crassinervis* and *D. chabrias*, which were not very well differentiated. The PCA found similar results to the CVA (Figure 3C,D). When geometric morphometrics are difficult to distinguish, we can combine them with barcode data.

#### 3.1.3. Phylogenetic Analysis

All three species of *Pachyneuron* were clustered on distinct branches, and the results demonstrated that the COI gene could differentiate the three species effectively. The results indicate that the intraspecific distances were 0% to 0.01%, and the interspecific distances between these three species varied from 0.12% to 0.18%. The bootstrap values of all three species were more than 70, proving that the evolutionary tree was highly credible. In *Diglyphus*, the results showed that the intraspecific distances were 0% to 0.03%, and the interspecific distances between these five species varied from 0.01% to 0.16%. However, *D. chabrias* and *D. crassinervis* were found to be genetically inseparable. The other species clustered together, and phylogenetic analysis provided favorable support for morphological identification, which was also consistent with this study. ABGD using the K80 Kimura measure of distance divided *Pachyneuron* into three groups, consistent with the ML and morphologic results. But ABGD failed to identify barcoding gaps within all available COI genes of *Diglyphus.* It was only divided into four groups; *D. chabrias* and *D. crassinervis* were not separated. The ML tree, traditional morphologic classification and GM were all in agreement, and the follow-up can be based on these results (Figure 4). All obtained sequences were uploaded to the GenBank database (Appendix A).

### 3.2. Studying Variations in Shape of Diglyphus isaea at Different Altitudes

A total of 125 female specimens were collected, and after removing specimens with body and wing mutilations, 102 photos of the right forewings and 89 photographs of the thorax of competitors were obtained for the GMA. When the PCA was applied to the 3 landmarks and 40 semi-landmarks of the forewing, the first two principal components accounted for 43.56% and 29.12% of the variance, and 72.68% of the total shape variance was accounted for by the cumulative variation. The 102 populations were clustered into three groups based on forewing shape, but there were also some areas of overlap, especially in the middle- and low-altitude areas, which indicates that the morphological variation is not significant. The wireframe of the PC1 positive tip of the posterior margin expands downwards and smv-mv widens upwards. The PC2 positive part of the forewing shows an inward movement of the wing margins (Figure 5A).

PCA was applied to the 30 semi-landmarks of the thorax. The first two principal components accounted for 26.71% and 23.50% of the total variation. The three-group had large overlapping areas, which indicates that the morphological differences between the middle and low elevations are not significant. In the wireframe, the PC1 positive area of the sidelobe of the mesoscutum widened and the PC2 positive area of the sidelobe and axilla extended. Analyzed as a whole, the thorax is shown to widen and the forewings are shown to become elongated at high altitudes (Figure 5B).

The CVA by geographical area for forewings and the thorax shows that the scatterplot analysis results of these populations can be divided into three groups (Figure 6). The high-altitude group is separated from the middle- and low-altitude groups, and CVA, as a discriminant analysis approach, identifies the elevational gradients in order to graphically visualize the shape variation between them. Based on the Mahalanobis and Procrustes distances among groups, the pairwise cross-validated reclassification test was efficient for separating the specimens. There are significant differences in the *p*-values (*p* < 0.0001) from the permutation tests for the Mahalanobis distances among groups, which indicates that the results of this study are statistically significant. Finally, CVA was performed for the forewings and thorax; the results can be clearly categorized into three clusters based on elevation. Firstly, there is an obvious reduction in the lowermost edge of the wing, suggesting that the wing surface is expanding. Secondly, based on the analysis of semi-landmarks, the primary area of change is at the wing’s end area, where it clearly expands, while the three selected landmarks remain almost the same.

### 3.3. Analysis of Shape Variability at Different Elevations in Pachyneuron aphidis

A total of 110 female specimens were collected. After removing specimens with body and wing mutilations, 87 photos of the right forewings and 91 photos of the thoraces were obtained. PCA analysis was applied to the 4 landmarks and 30 semi-landmarks of the forewing; the first two principal components accounted for 63.66%, 18.94% and 82.60% of the total shape variance. The results showed more pronounced changes in the wings, but there was still crossover (Figure 7A). For the 40 semi-landmarks of the thorax, the results of the scatterplot at higher elevations are clearly separated from those at middle and lower elevations. There is variation in thorax shape in the mesothorax (Figure 7B).

Thin-plate spline (Tps) deformation for PC1 showed that the distal margins of the wings exhibit significant expansion, while the proximal regions experience slight constriction. Additionally, there was an overall augmentation in the wing surface area (Figure 7C). Tps demonstrates that the changes in the thorax mostly occur in the mesothorax, where the wings are carried in the mesothorax and contract inward to become more elongated (Figure 7D).

The results of the CVA of the *P. aphidis* forewings and thorax showed more pronounced changes in the wings, which allowed for the division of the three groups, and there was no crossover. The species at higher elevations were clearly separated from those at middle and lower elevations. And there are significant differences in the *p*-values from permutation tests for Mahalanobis distances among groups (Figure 7E,F).

### 3.4. Analysis of Size Variation by Centroid Sizes

The centroid measurement was used to determine the size of the wings. The differences in the size of the wings and thorax were significant between the various altitudes. The results of this study showed that the centroid size (Log10CS) had a linear association with altitude. In the high-altitude areas, the forewing and thorax had larger centroid sizes than other populations. The CS of *D. isaea* in both the wing and thorax was positively correlated with altitude (*R* = 0.59, *p* < 0.001; *R* = 0.58, *p* < 0.001) (Figure 8A,B). In addition, *P. aphidis* also showed a positively correlated linear relationship (*R* = 0.29, *p* < 0.05; *R* = 0.3, *p* < 0.05) (Figure 8C,D). However, the trend in *P. aphidis* was not as significant as in *D. isaea.*

### 3.5. COI Gene Haplotype Analysis of D. isaea

#### 3.5.1. Analysis of Genetic Diversity

In this study, 44 COI sequences were obtained with a length of 713 bp. The genetic diversity of the *D. isaea* population was assessed through haplotype diversity *Hd* and nucleotide diversity *Pi*. When *Hd* > 0.5 and *Pi* > 0.005, the population showed high genetic diversity. Among the seven populations, the value of *Hd* ranged from 0.6667 to 1.0000 (*Hd* > 0.5). *Pi* ranged from 0.0001 to 0.0671; however, in the Altun mountain region, *Pi* < 0.005. The total *Pi* was 0.05452, and that of *Hd* was 0.96617. The results indicate that the population has high genetic diversity (Table 2).

#### 3.5.2. Haplotype Analysis of *D. isaea* Populations

In the present study, 44 COI gene sequences of *D. isaea* were obtained and categorized into seven geographic populations. The correlation between *Hd* and *Pi* and climatic factors and elevation was analyzed using the Mantel test. *Hd* and *Pi* have a significant correlation with longitude (r = 0.505, *p* < 0.05; r = 0.431, *p* < 0.05) (Figure 9A).

When populations are isolated by distance, an Fst value of 0 indicates no differentiation between subpopulations while 1 indicates complete differentiation. Only TSK, AKT and BL had low levels of genetic differentiation (Fst < 0). The genetic differentiation in the Altun Mountain National Nature Reserve population was zero. Secondly, the genetics and geographical distance were analyzed through the Mantel test (r = 0.423, *p* = 0.037 < 0.05). Among *D. isaea* populations, there is a significant positive correlation between genetic and geographic distances. The results illustrate that genetic differentiation among populations was influenced by geographic distance (Figure 9B).

We identified a total of 32 haplotypes. Of these, four were shared haplotypes and the remaining 28 (87.5%) were unique to a single population. The Pamir plateau (WQ and AKT) and the city of Bole did not share haplotypes. This suggests that some genetic differentiation had been exhibited among these geographic populations. The haplotype network diagram shows an overall star-shaped distribution, indicating that the population has undergone an expansion process. Haplotype distribution also showed a relationship that relatively corresponded to geographic distribution. For example, HM, ALMA and ALMW are located in the southeastern region of Xinjiang and clustered in the left part of the network diagram. BL is located in the northern region of Xinjiang and is clustered in the right part. However, numerous mountain ranges such as the Himalayas, Kunlun Mountains and Tianshan Mountains converge in the Pamir plateau; thus, the haplotypes were more complex (Figure 9C).

## 4. Discussions

### 4.1. Integrating Taxonomy to Define Diglyphus and Pachyneuron

In our study, the integrated taxonomic approach was used to classify species in the two genera, and the results suggest that the integrated method can be used to more comprehensively define closely related and smaller species [56]. In terms of the morphological classification of *Diglyphus*, the morphological characteristics of five species are very close, such as the color of the pedicle, the length-to-width ratio of the mesothorax, the size of the speculum and the color of leg segments. Nevertheless, *D. chabrias* and *D. crassinervis* are very close in pedicle and leg color, and the ratio of the mesothorax is close enough to be easily confused [57]. Through GM analysis, we used two methods, PCA and CVA. PCA was used to obtain preliminary observations of differences in wing morphology among groups, and no statistical methods were used to find differences between groups. CVA is a discriminant function to analysis method that allowed us to identify the groups and can also determine whether the selected characteristic values separate the groups [58,59,60]. However, the overlap areas between *D. chabrias* and *D. crassinervis* are relatively large and not easily delineated using geometric morphometric analyses, suggesting that the wing pattern differences between these two species are reduced. *Pachyneuron* shows a clear division of the three groups. PCA indicates that the forewing morphology of *P. aphidis* and *P. solitarium* is relatively similar, but using the seven landmarks in CVA allowed us to identify *Pachyneuron*. The Mahalanobis distances are clearly separate and interspecific.

DNA barcoding can resolve this difficulty; however, DNA barcoding gaps often exhibit a poorly defined divide. In the future, we will increase the number of genes and gene co-builders to classify species (COI and ITS2) [61]. As for the genus *Pachyneuron*, the species can be completely distinguished through morphological comparisons. For the two relatively similar species, *P. aphidis* and *P. solitarium*, GM and DNA barcoding can be used to classify them into different taxa. Therefore, these three approaches can be co-used in species delimitation. The parasitic wasp plays a significant role, and our research can provide reference data for subsequent classification studies of Eulophidae and Pteromalidae through GM [62,63,64].

### 4.2. Forewing and Thorax Shape Adaptation at Different Altitudes

Numerous studies have examined wing adaptations at different altitudes, and it is known that the environment can influence the body size of insects. *Drosophila melanogaster* and honeybees have been more intensively studied; researchers found that body size and wing shape can be disturbed by the external environment [65,66]. Our research used two kinds of parasitoid wasps and observed that variations in altitude had an impact on the shape and dimensions of their wings and thorax.

Insects’ wings act as a vital organ for flight and are essential for understanding how insects adjust to their environment. Current research focuses on three main areas: (1) genetic aspects [67]; (2) morphometric studies [68]; and (3) geometric morphometric analyses [69]. The present research utilizes GM analysis in the two genera. The results of CVA at three different altitudinal gradients showed that for the three groups, the morphology of the forewings changed significantly at different altitudinal gradients. The CS determined that forewing size was positively correlated with altitude. However, the R value was relatively low, and the R2 (coefficient of determination) is an important index of response goodness of fit. Therefore, the volume of specimens should be increased in subsequent studies to calculate R2. Throughout most of the life time of insects, the temperature during growth has the most important effect on body size, and bodies are usually bigger when temperatures are lower [70]. For instance, temperature was found to be a major natural factor in *Trilophidia annulata* (Orthoptera) and Zygoptera (Odonata) [71]. Similarly, Hymenoptera show this characteristic through the formation of larger wings as the altitude gradient increases [72,73,74,75]. In summary, the results of this research demonstrate that the morphology of the wings of parasitic wasps changes with altitude.

In parasitic wasps, the thorax is also a relatively important part of the body, and the wings are borne on the mesothorax [76]. Hence, our objective was to understand the difference in the thorax. The present research is the first attempt to use geometric morphometric studies of the thorax to investigate its shape adaptation to high altitudes. We found that the thorax of both species changed in morphology at different altitudes. The main morphological changes were in the middle of the axilla and mesoscutum, with proximity to forewing insertion. The tps deformation grids at the extremes show that the shape changes by contracting inwards. The rapid movement of the wings in small insects is powered by the indirect flight muscles. The dorsal longitudinal muscle is associated with an insect’s ability to fly, and both diastole and muscle contractions pull on the dorsal plate of the mesothorax, which is a longitudinal muscle [77,78,79,80]. Larger wings and a greater flight ability at higher altitudes require more muscle and may result in a narrower and more elongated mesoscutum. In the future we intend to use micro-CT techniques to reconstruct the muscle composition of the mesothorax in order to evaluate our conjectures [81,82].

### 4.3. Utility of the COI Gene in Population Genetic Differentiation

Mitochondrial DNA (mtDNA) is widely used in studies of insect genetic diversity, genetic differentiation, population dynamics, dispersal patterns and phylogeny [83]. Genetic differentiation refers to the degree of variation in genetic traits among individuals within a population and can be explained by the number of shared haplotypes, the haplotype network, Fst and gene flow values. In this research, we found 4 shared haplotypes and 28 single haplotypes. The two cases of haplotype co-existence indicate that the *D. isaea* population not only retains the original and adaptive shared haplotypes but also that populations have developed unique haplotypes adapted to their local environments. *Hd* and *Pi* were found to have significant correlation with longitude, which may be due to changes in geographic location and vegetation diversity, with more mountain ranges at lower longitudes and more plains at higher longitudes, leading to an increase in *Hd* and *Pi*. Researchers have found that altitude is an important environmental factor influencing genetic diversity. However, elevation did not show a correlation in this study, presumably due to the small number of specimens at different elevation gradients. In the future, we will increase the number of specimens to verify the results. The positive correlation between Fst and geographic distance suggests that genetic differentiation between populations is affected by geographic distance.

High-altitude environments are characterized by a variety of complex environmental factors such as strong ultraviolet radiation and low temperatures, oxygen and atmospheric pressure [84], which may lead to morphological changes. In our results, we found a positive correlation between the size of the parasitoid forewings and thorax and altitude. Furthermore, genetic differences were observed in seven geographic regions at the species level, showing a high level of genetic differentiation and a significant correlation with longitude. Body shape and genetic diversity preliminary confirmed two important factors in the adaptation of parasitic wasps to extreme environments. Furthermore, it also provides reference data for the study on the adaptation of parasitoid wasps in high-altitude regions. In the future, we will use genomics, such as RNA-seq, to obtain signaling pathways or genes associated with body shape changes, as well as parasitological research to determine the parasitic potential. Eventually, we will conduct a more in-depth investigation of parasitic wasps in Xinjiang.

## Figures and Tables

**Figure 1 insects-15-00497-f001:**
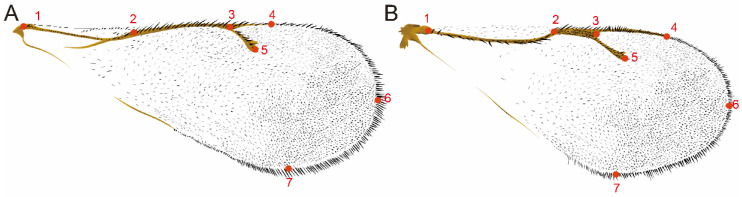
Geometric morphometric landmarks in the forewing: (**A**) *Diglyphus*; (**B**) *Pachyneuron*. The numbers 1–7 represent the locations of the seven landmarks: 1. the beginning of submarginal vein; 2. the ending of submarginal vein; 3. the ending of the marginal vein; 4. the ending of the post marginal vein; 5. the ending of the stigma vein; 6. the tip of the fore wing; 7. the tip of the posterior margin of the forewing; 1–2. submarginal vein (smv); 2–3. marginal vein (mv); 3–4. post-marginal vein (pmv); 3–5. stigma vein (stv).

**Figure 2 insects-15-00497-f002:**
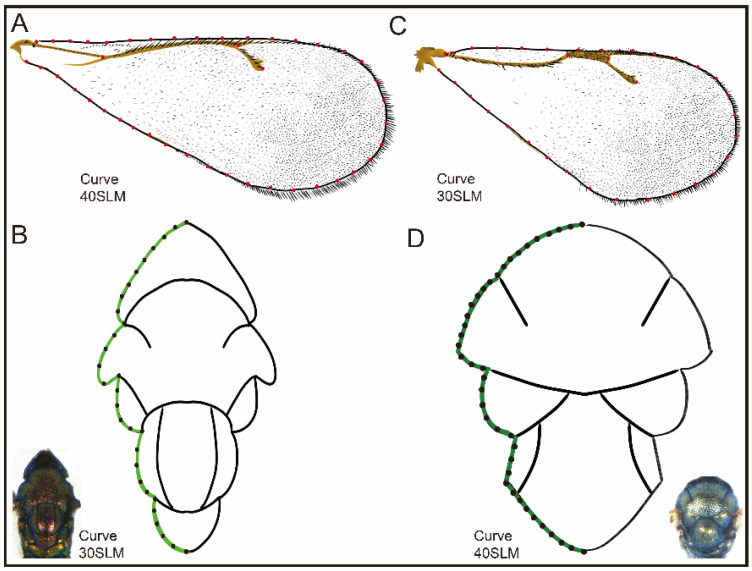
Landmarks and curves of geometric morphometrics in forewing and thorax: (**A**) *D. isaea* forewing; (**B**) *D. isaea* thorax; (**C**) *P. aphidis* forewing; (**D**) *P. aphidis* thorax.

**Figure 3 insects-15-00497-f003:**
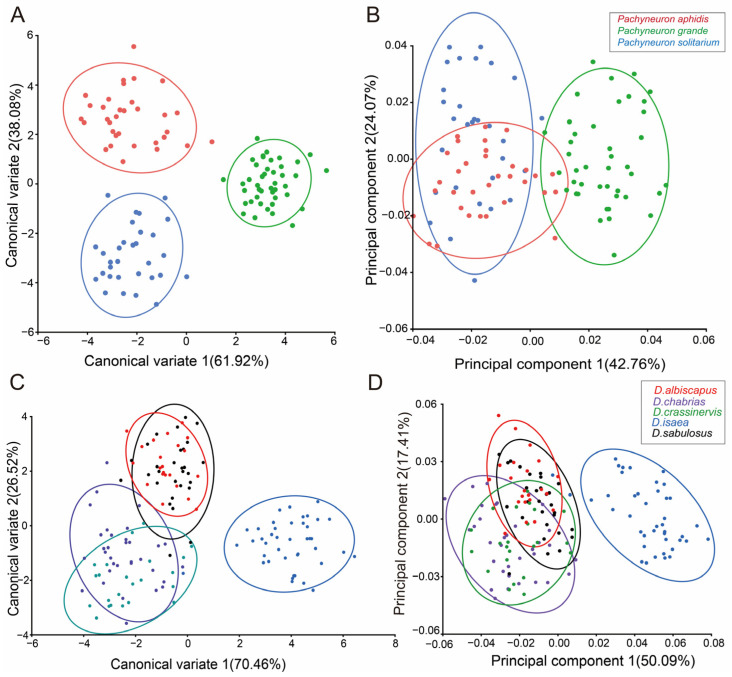
GMA of genus *Pachyneuron* and *Diglyphus* forewings. (**A**) CVA for *Pachyneuron* forewings; (**B**) PCA for *Pachyneuron* forewings; (**C**) CVA for *Diglyphus* forewings; (**D**) PCA for *Diglyphus* forewings. The percentages in the graph indicate variation rates, while the ellipses reflect 90% confidence intervals around each data cluster.

**Figure 4 insects-15-00497-f004:**
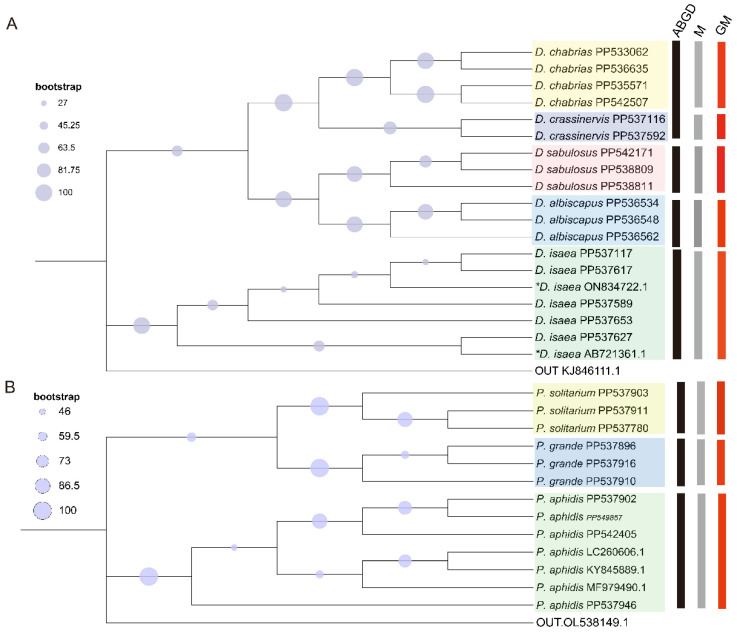
A maximum likelihood phylogeny tree was constructed to reveal the relationships among the *Diglyphus* five species and *Pachyneuron* three species using a combined molecular analysis of mt DNA COI: (**A**) sequences of COI from five species of *Diglyphus*, with *N. tutae* (KJ846111.1) as an outgroup; (**B**) relationships among the sequences of COI from three species of *Pachyneuron*, with the *H. longipetiolus* (OL538149.1) as an outgroup. * Sequences were downloaded from the NCBI database, and the purple circle represents bootstrap. M indicates morphological classification.

**Figure 5 insects-15-00497-f005:**
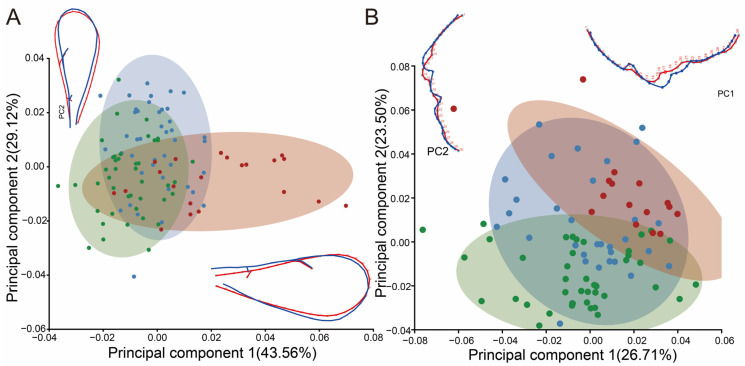
GM analysis of *D. isaea* forewings. (**A**) PCA for *D. isaea* forewings; (**B**) PCA for *D. isaea* thorax. The red point represents high (about 3000 m), green represents middle (about 2000 m) and blue represents low (about 800 m) altitude. The initial shape is shown by red wireframes.

**Figure 6 insects-15-00497-f006:**
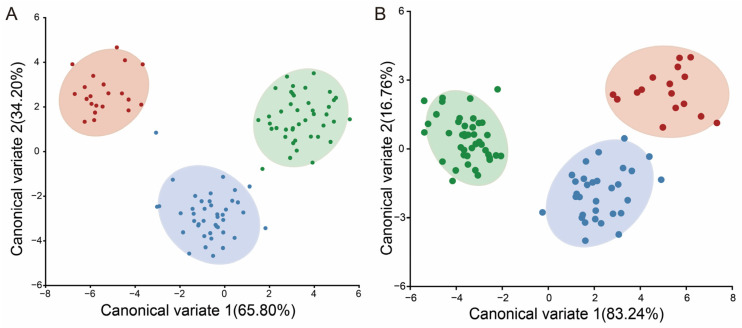
GM analysis of *D. isaea* forewings: CVA of *D. isaea* (**A**) forewings and (**B**) thorax. The red represents high (about 3000 m), green represents middle (about 2000 m) and blue represents low (about 800 m) altitude.

**Figure 7 insects-15-00497-f007:**
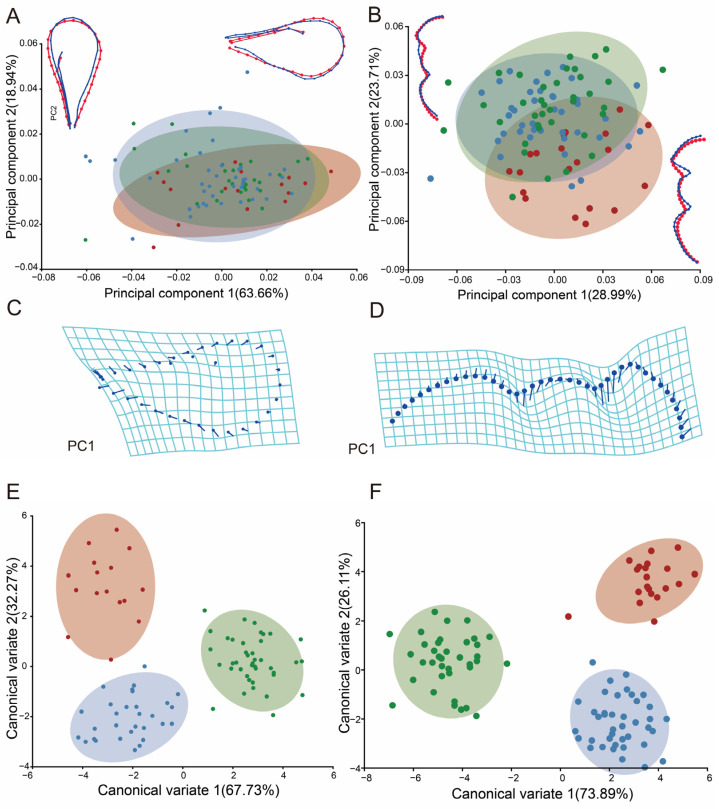
GM analysis of *P. aphidis* forewings and thorax: (**A**) PCA for *P. aphidis* forewings; (**B**) PCA for *P. aphidis* thorax; the initial shape is shown by red wireframes. TPS deformation grids at the extremes of each axis show the shape change represented by (**C**) the associated forewing PC1 and (**D**) the associated thorax PC1. CVA of *P. aphidis* (**E**) forewings and (**F**) thorax. The red point represents high (about 2000 m), blue represents low (about 500 m) and green represents middle (about 1000 m) altitude.

**Figure 8 insects-15-00497-f008:**
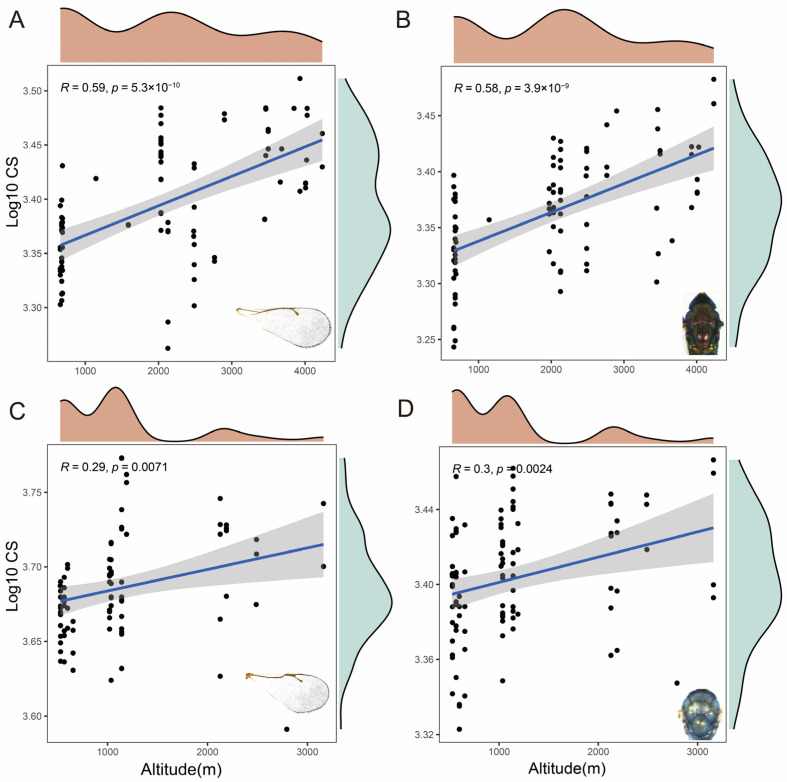
Variation in body size at different elevations of two species of parasitic wasps: CS of (**A**,**B**) *D. isaea* and (**C**,**D**) *P. aphidis* forewings and thorax at different altitudes. The vertical coordinate is log10(cs); *p* < 0.05 indicates a significant difference and *p* < 0.001 indicates a highly significant difference, respectively. *R* indicates the Pearson correlation coefficient.

**Figure 9 insects-15-00497-f009:**
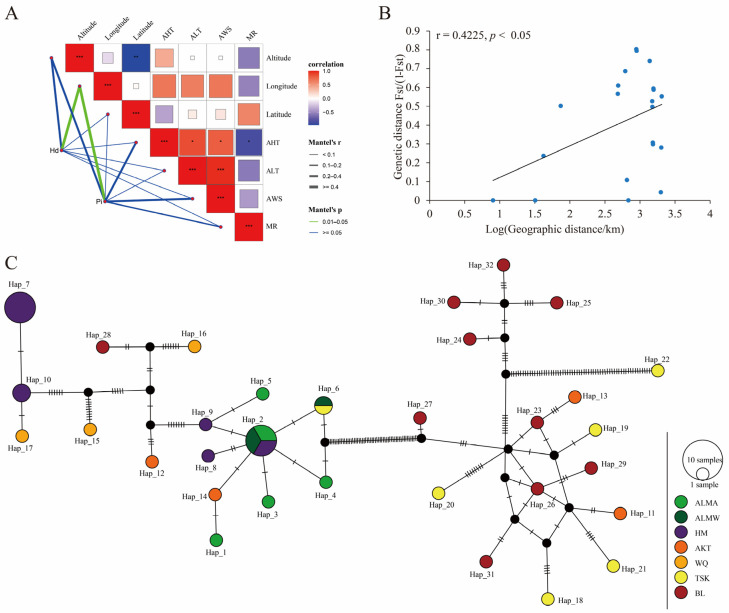
Genetic diversity analysis and haplotype network. (**A**) Mantel test between environmental factors and *Hd* and *Pi*; (**B**) the relationship between genetic and geographic distances based on Mantel tests; (**C**) median-joining network based on the COI gene haplotypes. Each circle represents a haplotype, and the area of the circle is proportional to the number of specimens with that haplotype. Colors within nodes refer to *D. isaea* sampling regions. * *p* < 0.05, ** *p* <0.01, *** *p* < 0.001.

**Table 1 insects-15-00497-t001:** Information about sampling sites for *Diglyphus* and *Pachyneuron* populations from Xinjiang.

Region	Location	Coordinates	Altitude	Number
Pamir plateau	Aketo county	38°53′54″ 75°12′5″	2764 m	55
Pamir plateau	Ulugqat county	39°52′17″ 75°32′41″	2127 m	40
Pamir plateau	Artush city	39°39′40″ 75°31′23″	2899 m	35
Pamir plateau	Tashkurgan county	37°42′24″ 75°17′30″	3001 m	48
Altun mountain	Ruoqiang county—1	37°58′15″ 88°57′5″	3930 m	23
Altun mountain	Ruoqiang county—2	37°48′14″ 89°54′41″	3449 m	18
Altun mountain	Ruoqiang county—3	37°39′45″ 89°48′26″	3941 m	22
Basin	Hami city	42°47′58″ 93°21′9″	679 m	72
Plain	Emin county	46°37′49″ 83°48′30″	567 m	85
Plain	Habahe county	48°03′36″ 86°23′10″	530 m	48
Basin	Shache county	38°25′10″ 77°7′32″	980 m	61

**Table 2 insects-15-00497-t002:** Haplotype diversity nucleotide diversity and different geographic populations of *D. isaea.*

Region	Population	h	*Hd*	*Pi*	K
Ruoqiang county (Altun mountain)	ALMA	5	0.9333	0.0028	2.0000
	ALMW	2	0.6667	0.0001	0.6667
Hami city	HM	5	0.7424	0.0134	9.5303
Kashi region (Pamir plateau)	AKT	4	1.0000	0.0671	47.8333
	WQ	3	1.0000	0.2338	16.6667
	TSK	6	1.0000	0.0624	44.4667
Bole city	BL	10	1.0000	0.0321	22.8889

Note: h indicates number of haplotypes; *Hd*, haplotype diversity; *Pi*, nucleotide diversity; K, average number of differences.

## Data Availability

All other data presented in this study are available within this manuscript and its additional files. DNA barcode data are uploaded to the Genbank database.

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
