# Peer review of "Geometric Morphometrics and Genetic Diversity Analysis of Chalcidoidea (Diglyphus and Pachyneuron) at Various Elevations"

_insects, 2024, doi:10.3390/insects15070497_

Round 1

Reviewer 1 Report

Comments and Suggestions for Authors

This study reports species diversity and using morphometric and COI barcoding to explore diversity of parasitic wasp species in Xinjiang, China. Following are my comments and suggestions:

- title, I think it is incomplete sentence, I suggested replace “two genera” with “parasitic wasps”.

- L15, what is means by “preliminary illustration”?

- L23, “analysis” not “Analysis”

- L41, remove “etc”

- L67, replace “periods” with “stages”

- L69, remove “in production”

- L72, replace “fauna” with “regions”

- L85, how high? provide information of the elevation range of this region.

- L86, remove “etc”

- L113, section 2.2 should be “Morphological analysis” and section 2.2.3 should separate to another section i.e. 2.3 under subtitle “DNA barcode” or “Molecular analysis”. Section 2.2.1 should be “morphological identification” and 2.2.2 should be “morphometric analysis”. The section 2.3 should place under the section 2.2.2. Section 2.4 should be under section “molecular analysis”.

- Section 2.2.3, I suggest that in addition to the comparisons of the COI sequences only between those obtained in the present study, these sequences should analyze using identification engine function in BOLD (https://www.boldsystems.org/index.php/IDS_OpenIdEngine) for comparisons with those other reported from other geographic regions (if they are existing in this database).

- Results, again, I suggested that section 3.1 should be “morphological identification”

- L251-257, this part is data analysis therefore, should be moved to Materials and Method section. In addition, all accession numbers of sequences obtained in present study and those retrieved from GenBank must be provided.

- Figure 3, 5 and 6, it will be useful to provide the main contribution parameters in X and Y axes of the scatter plots.

- L264, “relatively close but cannot be separated.” should be “genetically inseparable.”

- Table 3, base compositions are not important result and can be omit because COI gene is commonly report in several organisms.

- L253 – 254, R values are relatively low although statistically significant. It is more accurate to report as R2 which will be lower than the R. indicate very weak relationships. Please acknowledge this limitation in discussion.

- L418, but DNA barcode cannot separate D. chabrias and D. crassinervis (Fig. 4). Please discuss this limitation.

- L483, again remove “etc”

- L484, “two” what?

- L486, is there any evidence that function of the COI gene is related to morphological characters examined in the present study? If so, please provided otherwise the unique COI haplotypes is possibly not related to morphological adaption in this study.

- L495 – 497, there is an exception, please see above comments, not all morphological species were molecularly separated.

- English language need improvement.

Author Response

Dear reviewer: Thank you very much for taking the time to review our manuscript.                          Please find the detailed responses below review report. 

Reviewer 2 Report

Comments and Suggestions for Authors

In the reviewed MS the authors presented the novel data on morphological and genetic variability of wasps collected in China. They collected members of two chalcid wasp genera, investigated their using traditional morphological methods. They also investigated the correleation between geography (altitude) and the shape of forewing and body using geometric morphometrics. Finally, they obtained sequences of COX1 gene and compared congruency of the results obtained with different morphological and molecular species delimitation methods. The MS is written in poor English language. The Introduction is slightly unfocused, the goals of the study are indicated indistinctly. The title of the MS and ther Abstract need revision. In general, the title should prcisely correspond to the results, obtained in the study. Some images needs revisions. The authors are requested to explain better the results of statistical analyses, specifically the discongruence of the results of principal component and canonical variant analyses. The discussion is too long. Some additional remarks are below.

Title: the title is suboptimal. Please make it coherent with the content of the last paragraph of the Introduction (lines 91-99) where you specify the goals of the study. Do not overgeneralize, please mention the genera which you studied in the title (Diglyphus and Pachyneuron)

15, 17: do you really need “preliminary” here? Preliminary results are ok for a conference, but not for a paper

21: “homozygous” – it is unclear, why do you use this term here?

23: “Then, Analysis” à analysis

The content of the Absract clearly indicate that this study reports “intermediate/preliminary” results.

55-64: This paragraph seems redundant

74: [24, 25]. some à Some

86: “It provides us with ideas for investigating” --- suboptimal wording

91-99: the aim and goals of the study are described indistinctly, Please, revise this paragraph.

148: Please, explain why some letters are in brackets in the primer sequence - COIS-F: 5’-TAAGATTTTGATTATT(AG)CC(TA)CC-3’

160: “A total of 32 COI gene sequences were acquired. Of these, 19 sequences belong to 160

Diglyphus, while 13 to Pachyneuron.” --- please, specify here how many different morphospecies were covered by sequencing

167: please specify detail of the ML analysis. Which model was used? How the ML tree was estimated etc.

198: mantel à Mantel, please, provide a reference with the Mantel test

223: Section 3.1.2, this is very confusing that principal component and canonical variate analyses resulted in so different plots (Fig.3A and 3B). This needs careful explanation and recalculation. Additionally, this section needs major linguistic revision / rewording.

252: “Two sequences, Necremnus tutae (Eulophidae) and Halticoptera longipetiolus (Pteromalidae) were downloaded from NCBI as outgroups. The examination of the ML tree was conducted using bootstrap with 1000 replicates.” --- this data should be given in the section Material and Methods

272: Table 2 could be given in the Supplement

275:  Please, root the trees. Please, explain why did not you perform a combined analysis, but present two different trees?

374: D. isaea--- italic

423-428: this paragraph seems redundant, because it does not report any important data/conclusion

429: sections 4.2. and 4.3 are too long and somewhat repetitive. Please consider making them shorter and combining into a single section

495-503: this paragraph needs rewriting.

Comments on the Quality of English Language

Extensive editing of English language required

Author Response

Dear  reviewer: Thank you very much for taking the time to review my manuscript.                   The main issues we've answered that in the "Response to Reviewer". 

Round 2

Reviewer 1 Report

Comments and Suggestions for Authors

Dear Author,

Thank you for your revised manuscript. It is now much improve. Also, thank you for considering my comments and suggestions and revised the manuscript accordingly.

Author Response

Dear reviewer:

          Thanks for the advice on our manuscript, we learned a lot too. Thanks again.

Reviewer 2 Report

Comments and Suggestions for Authors

The MS became better after the revision, but it still needs some improvements

1. The title is suboptimal. It is recommended to avoid the term "adaptation" because in this paper the authors do not provide any evidence that the morphometric and genetic differences which they detected are true adaptation. Therefore, this is only "speculation" or "hypothesis".

2. When the authors mention various software and algorithms (iqtree, bootstrap, modelFinder, ABGD etc), they should cite the appropriate papers. Mentioning Phylosuite is not enough, Phylosuite is just a frame. Please, see papers on iqtree by Minh et al. and look how to cite it correctly.

3. line 498 Body shape and genetic diversity confirmed the special evolutionary status of parasitic wasps.--- it is not clear what exactly do you mean, saying this. Could you be more precise?

4. line 499. In the future, we will use genomic and parasitological research to perform a more in-depth investigation of parasitic wasps in Xinjiang. --- please specify more distinctly the perspective: what exactly do you need to investigate using these methods and how is it connected with the results obtained in this study.

5. The Abstract needs to be written more smooth.

6. Please specify high level taxonomy of the wasps in the title. Probably you could avoid "parasitic" in the Title

Comments on the Quality of English Language

minor

Author Response

Dear reviewer:

            Thanks for the advice on our manuscript, we have revised it again, thanks again. The specific changes are in the word file.
